# Exploring the Rumen and Cecum Microbial Community from Fetus to Adulthood in Goat

**DOI:** 10.3390/ani10091639

**Published:** 2020-09-11

**Authors:** Xian Zou, Guangbin Liu, Fanming Meng, Linjun Hong, Yaokun Li, Zhiquan Lian, Zhenwei Yang, Chenglong Luo, Dewu Liu

**Affiliations:** 1College of Animal Science, South China Agricultural University, Guangzhou 510640, China; zouxian08@163.com (X.Z.); gbliu@scau.edu.cn (G.L.); linjun.hong@scau.edu.cn (L.H.); ykli@scau.edu.cn (Y.L.); 18820425216@163.com (Z.L.); zivir.y@aliyun.com (Z.Y.); 2Institute of Animal Science, Guangdong Academy of Agricultural Sciences & State Key Laboratory of Livestock and Poultry Breeding & Guangdong Key Laboratory of Animal Breeding and Nutrition, Guangzhou 510640, China; mengfanming@gdaas.cn

**Keywords:** goat, fetus, kid, microbiome, rumen, cecum

## Abstract

**Simple Summary:**

The rumen and cecum are two important fermentation organs in ruminants. The acquisition and development of the neonatal microbiome, as well as the difference between these two organs, was important. We performed 16S rRNA gene sequencing to study the diversity, structure and composition of the goat microbial communities between the rumen and cecum from fetus to adulthood. The results revealed the microbial transmission routes from the mother to fetus, and also revealed the establishment and dynamic fluctuations of the gut microbiome from fetus to adulthood in goats.

**Abstract:**

The present study aimed to investigate the colonization process of epithelial bacteria attached to the rumen and intestinal tract tissue during the development of goats after birth. However, this process from fetus to adulthood was very limited. In goats, the rumen and cecum are two important fermentation organs, and it is important to study the acquisition and development of the neonatal microbiome, as well as the difference between these two organs. To characterize the microbial establishment and dynamic changes in the rumen and cecum from fetus to adulthood, we performed 16S rRNA gene sequencing for 106 samples from 47 individuals of nine pregnant mother–fetus pairs and 16 kids from birth up to 6 months. The diversity, structure and composition of the microbial communities were distinct between the rumen and cecum after birth, while they were similar in the fetal period. The study showed a rapid loss and influx of microbes at birth, followed by slight selection after drinking colostrum, and then a strong selection after weaning, suggesting that the establishment and dynamic fluctuations of the gut microbiome undergoes three distinct phases of microbiome progression in life: a conserved phase (during late pregnancy in the fetus), a transitional phase (newborn until weaning), and a stable phase (from weaning to adulthood). The results supported the view that microbes exist in the fetus, and revealed the establishment and dynamic fluctuations of the gut microbiome from fetus to adulthood in goats.

## 1. Introduction

The rumen is the organ wherein plant digestion enables the conversion of plant fibers into chemical compounds (acetate, propionate and butyrate), which are subsequently absorbed and digested by the animal [1]. During the suckling milk period, the rumen is inactive and is not involved in the digestion of plant material. The changes in the structural and physiological properties of the rumen with age are linked to the development of rumen microorganisms. The fermentation products of rumen microorganisms which are responsible for the absorption of nutritional components play important role for the development of the rumen wall villi [2,3,4]. The cecum is also an important fermentation organ in ruminants. Most of the chyme that has not been fermented completely in the rumen, then will be fermented to produce final metabolites in the cecum and other areas of the hindgut, such as short-chain fatty acids (SCFAs), which are absorbed and utilized in the cecum [5]. Several studies have reported that the bacterial communities in the gastrointestinal tract (GIT) are not only influenced by diet, but also by the age of the animal [6,7,8,9]. In cows, the bacterial composition of rumen content undergoes dynamic changes during the first 2 years, and the levels of members of the phylum Bacteroidetes and Firmicutes increased with age, while those of Proteobacteria decreased with age [6]. In goats, microbial colonization of the rumen contents occurs at 1 month, functional achievement at 2 months, and anatomic development after 2 months [10]. The colonization process of the microbiota along different GIT compartments in kids (Capra aegagrus hircus) (0, 14, 28, 42 and 56 days old) could be divided into three stages: initial phase, transit phase, and the relatively stable phase [9]. However, this process from fetus to adult is poorly understood.

The role and importance of prenatal microbial colonization and the presence of intra-amniotic microbiota are still a matter of debate; however, they have been demonstrated in several vertebrates, such as mice, chicken and rock pigeon [11,12,13,14]. In humans, that an infant is sterile at birth has been challenged by an increasing number of studies offering evidence of the high microbial diversity and strain heterogeneity in the meconium of newborns [15,16,17,18]. In addition, the detection of bacterial DNA in the placenta, amniotic fluid and umbilical cord blood gave rise to the hypothesis that the human fetus might be exposed to microbiota in utero through the ingestion of amniotic fluid [13,18,19,20,21]. Hence, we hypothesized that microbes exist in the fetus and umbilical cord blood.

The acquisition and development of the neonatal microbiome is key to establishing a healthy host–microbiome symbiosis, and influences the productive efficiency of the mature animal. In bovine and goat, some rumen bacteria that are essential for mature rumen function could be detected as early as 1 day after birth, long before the rumen is active or even before the ingestion of plant material occurs [6,7,10]. The rumen and cecum are two important fermentation organs in ruminants. In this study, we performed 16S rRNA gene sequencing to analyze and compare the microbial composition, abundance and dynamic distribution for 106 samples from 47 individuals of nine pregnant mother–fetus pairs and 16 kids from birth up to 6 months in goats. We aimed to characterize the gut microbial establishment and dynamic changes from fetus to adulthood in goats.

## 2. Materials and Methods

### 2.1. Ethics Approval and Consent to Participate

All experimental procedures and sample collections were conducted in accordance with the Regulations for the Administration of Affairs Concerning Experimental Animals (Ministry of Science and Technology, China; revised in August 2011) and the NIH Guide for the Care and Use of Laboratory Animals. The protocol for the animal trial was approved by the Institutional Animal Care and Use Committee of the South China Agricultural University (Approval No. 2018-G001), Guangzhou, China. All the goats received freely available water and were fed total mixed rations (TMR) (purchased from Wens Foodstuff Group Co., Ltd., Yunfu City, China) throughout the experiment. All surgery was performed under anesthesia with isoflurane or propofol, and all effort was made to reduce the number of animals used and minimize animal suffering.

### 2.2. Study Design and Animal Sampling

In the present study, Leizhou goats were obtained from the South China Agriculture University, Guangdong, China. Nine healthy pregnant goats aged 2.5 to 3.2 years and their 22 fetuses were used in this study. Goats were estrous synchronized before breeding and pregnancy was confirmed using ultrasound at 40 days post-breeding. The nine goats were pregnant at three time points: at approximately 90, 100 and 120 days, respectively, carrying a singleton, twins or triplets, including 10 females and 12 males (Appendix A). The pregnant goats were anesthetized and the fetus extracted and then euthanized. In addition, 16 of the kids were slaughtered and sampled for rumen contents and cecum contents at four time points: within one hour from birth (newborn) without receiving breast milk, 24 h after receiving breast milk, three months old, and six months old after weaning (Appendix A). By the respective deadline, all goats were sacrificed without prior feeding. All goats were slaughtered in two commercial abattoirs at different dates, and samples were collected at the slaughterhouse (Appendix A). For the pregnant goats, the three-month-old kids and the six-month-old kids, the enterocoel was opened and the rumen and cecum were separated with a suture line to avoid reflux of the digesta among adjacent regions after slaughter. Then, the cecal contents were collected into 2 mL tubes using sterile pipette tips within 10 min. In addition, a quantity of the rumen contents was tightly squeezed through four layers of cheesecloth into a vessel continually sparged with CO_2_ to yield ruminal fluid and squeezed solids, and the fluid portion (approximately 50 mL) was immediately placed on ice in a sealed tube as previously described [22]. Samples of umbilical cord blood and amniotic fluid were collected from pregnant goats within 20 min. The whole placenta was separated and put into a plastic tray with disposable plastic bags. Amniotic fluid (approximately 50 mL) was aspirated using a sterile syringe by passing a sterile needle through the amniotic membrane, and immediately placed on dry ice. Umbilical cord blood was collected by gently squeezing into 2 mL tubes, and was then snap-frozen in liquid nitrogen. The rumen fluid samples (approximately 10 mL) from the fetus were taken using a sterile needle and immediately placed on ice, and the contents of the ceca were collected by gently squeezing the digesta into 2 mL tubes and were then snap-frozen in liquid nitrogen. The rumen contents were obtained through centrifugation at 1000× *g* for 15 min at 4 °C. The centrifugal precipitation was then placed into 2 mL tubes and snap-frozen in liquid nitrogen.

The sampling process was performed by researchers wearing facial masks and sterile gloves and using a sterile scalpel and instruments. All samples were stored at −80 °C until DNA extraction.

### 2.3. Sequencing and Sequence Analysis

Bacterial DNA was extracted using an Omega E.Z.N.A. Soil DNA Kit (Omega Bio-tek, Inc., Norcross, GA, USA) following the manufacturer’s instructions, including three kit blanks (DNA extraction performed with ultrapure water addition). The V4 and V5 hypervariable region of the 16S rRNA gene was amplified by PCR using the sample-specific sequence barcoded fusion primers forward 5′-GTGCCAGCMGCCGCGGTAA-3′ and reverse 5′-CCGTCAATTCMTTTRAGTTT-3′. The PCR reaction conditions were as follows: 94 °C for 5 min; 94 °C for 30 s; 55 °C for 30 s; and 72 °C for extension. This was repeated for 27 cycles and a final 72 °C for 5 min. The PCR products were excised from a 1.5% agarose gel and purified using an AxyPrep DNA Gel Extraction Kit (Axygen Biosciences, Union City, CA, USA). In addition, all sample sequencing runs included three negative controls (in the form of a PCR-amplified kit blank sample). Purified PCR products from 109 samples were used to construct a sequencing library using Illumina TruSeq (Illumina, San Diego, CA, USA) following the manufacturer’s suggested protocols.

Data processing was performed using Quantitative Insights into Microbial Ecology (QIIME 1.9.0) [23]. Barcoded V4 and V5 amplicons were sequenced using the pair-end method by Illumina Miseq. Original pair-end sequences with a mean quality lower than 30, containing ambiguous bases, a sequence length shorter than 150 bp, chimeras, adaptor contamination or host contaminating reads were removed. The original pair-end sequence reads that passed our quality control criteria and contained a sequence overlap of at least 10 bp without any mismatch were assembled according to their overlapping sequences. The sequence analysis, including OTU (Operational Taxonomic Unit) clustering and taxonomy assignment, was performed in Mothur [24]. The Mothur standard operating procedure for MiSeq data was followed with the exception that potential contaminant genotypes (16S rDNA sequences possibly originating from reagents or instruments) were removed from the data after preclustering and chimera removal. The filtering logic was based on comparing the shared OTUs in samples and negative controls (in the form of a PCR-amplified kit blank sample), and OTUs shared with the negative controls were discarded.

In addition, negative controls were used to check for anomalous amounts of specific OTUs as a way to control for contamination. Based on the amount of sequences present in the negative control, the amounts of said sequences present in the biological samples and published material on contamination [25], a decision was made to remove all OTUs representing the genera Ralstonia, Ochrobactrum and Psychrobacter from the data sets. After removal of the suspected contaminants, a total of 2,959,806 sequences from the V4 and V5 region of the 16S rRNA sequence from 106 samples that passed our quality filters were used for this study. Trimmed sequences were uploaded to QIIME for further analysis. This type of data decontamination minimizes false positive observations, but will delete some taxa that were genuinely present in the samples.

### 2.4. Taxonomic Classification and Statistical Analyses

Taxon-dependent analysis was conducted using the Greengenes database [26]. Operational taxonomic units (OTUs) were counted for each sample to express the richness of bacterial species with an identity cutoff of 97%. Low-abundance OTUs (fewer than five reads) were filtered out of our analysis [27]. An analysis of microbial community diversity at different levels (phylum, class, order, family and genus) was performed using the Ribosomal Database Project (RDP) Classifier by comparing sequences to the Silva 123 database (Appendix A). Beta diversity was visualized using principal component analysis (PCoA).

### 2.5. Prediction of Microbial Function

The microbial functional profile was predicted using a phylogenetic investigation of communities via reconstruction of unobserved states (PICRUSt) [28]. The OTU abundance was normalized automatically using 16S rRNA gene copy numbers from known bacterial genomes in Integrated Microbial Genomes (IMG). The predicted genes and their functions were precalculated in the database of the Kyoto Encyclopedia of Genes and Genomes (KEGG) database, and the differences among groups were compared using the software STAMP. A two-sided Welch’s t-test and the Benjamini–Hochberg false discovery rate (FDR) correction were used in the group analysis.

### 2.6. Statistical Analysis

One-way analysis of variance was conducted for the differences in the alpha diversity, the variance of relative abundance of bacterial taxa, the total copy number of bacterial 16S rDNA gene, and the relative abundance values of KEGG pathways using the JMP 8.0 software (SAS Institute, Cary, NC, USA). Correlations between bacterial taxa and KEGG pathways were evaluated by Spearman’s correlation test using the R “heatmap” package. All data were expressed as the mean ± standard error, and significance was determined as *p* ≤ 0.05.

## 3. Results

### 3.1. 16S rRNA Gene Sequencing to Study the Establishment and Dynamic Fluctuations of the Goat Gut Microbiota

To analyze the composition and diversity of the microbiota, we performed 16S rRNA gene sequencing for all 106 samples from 47 individuals (Figure 1, Appendix A). Samples from nine healthy pregnant goats and their 22 fetuses were analyzed. The pregnant goats’ rumen content, cecum content, umbilical cord blood, amniotic fluid content and their fetus’ rumen fluid and cecum content were sampled after slaughter. Accordingly, we collected the rumen and cecum contents from 16 kids to elucidate the dynamic fluctuations of rumen and gut microbiota (newborns without receiving breast milk (NB, *n* = 4), one-day-old after receiving breast milk (L1d, *n* = 4), three-month-old kids after weaning (L3m, *n* = 3), and six-month-old kids (L6m, *n* = 5) (Figure 1B, Appendix A)).


### 3.2. Contamination Control and Removal of Potential Contaminant Sequences from 16S rDNA Amplicon Sequencing Data

Negative controls, in the form of a PCR-amplified kit blank sample from which DNA extraction was performed with ultrapure water addition, were used to check for anomalous amounts of specific OTUs as a way to control for contamination because of DNA extraction and PCR reagents, which always contain small amounts of bacterial DNA. PCoA clustering analysis based on 16S rRNA gene sequences showed that the bacterial microbiota from negative controls were distinct from the other samples (Figure 2A). The bacterial microbiota from the fetus and umbilical cord blood samples were distinct from the negative controls, and there were still many microbial species (e.g., Unclassified_Ruminococcaceae) in the fetus, even though we removed all OTUs representing the genera (the relative abundance >1% in FR or FC groups) from the data sets in the negative controls (Appendix A). As such, we hypothesized that microbes were present in the fetus and umbilical cord blood.

Compared with the negative controls, a portion of the reads represented genotypes from the fetuses and their mother’s samples that were shared with the negative controls (Appendix A). The result showed that only 38.25 ± 1.49% of sequence reads (average ± SD) represented genotypes in the fetus samples that were not detected in any of the negative controls, and 78.77 ± 4.11% in their mothers (orange bars in Figure 2B, Appendix A). However, some shared OTUs representing the families Clostridiales (order), Ruminococcaceae, Prevotellaceae and S24-7 were the major microbes in the healthy adult goat [29,30]; therefore, they should not be discarded as potential contaminants. Based on the amount of sequences present in the negative control, the amounts of said sequences that were present in the biological samples (published material on contamination [25]) and the major microbes in the healthy adult goat, a decision was made to remove all OTUs representing such genera as Ralstonia, Ochrobactrum and Psychrobacter from the data sets (50 genera in total, Appendix A). Thus, in the fetal samples, 30.32 ± 1.47% of sequence reads were discarded as potential contaminants, while only 1.08 ± 0.89% were discarded from their mothers’ samples (gray bars in Figure 2C, Appendix A). In addition, these sequence reads were also discarded as potential contaminants from the kid and umbilical cord blood samples (gray bars in Figure 2C, Appendix A). Finally, a total of 2,959,476 V4–V5 16S rRNA sequence reads from the 106 samples, with an average of 27,919 sequence reads for each sample (Appendix A), were used in this project. The total number of OTUs reached 3900.

### 3.3. Microbial Transmission from Fetus to Adulthood

To study the development of the microbiome from fetus to kid and kid to adult, we compared the microbiota of the rumen and cecum from 16 kids and the nine mother–fetus pairs. The ACE and Shannon indices differed significantly among the groups in the samples of rumen and cecum (*p* < 0.001) (Figure 3, Appendix A). The ACE value in the one-day-old group was lower compared with that in the other stage groups in the samples of rumen and cecum (*p* < 0.05) (Figure 3A). With regard to the bacterial diversity, the Shannon index increased first and then decreased at late pregnancy in both rumen and cecum samples, while it increased with increasing age after birth (except for the one-day-old cecum samples) (Figure 3B). PCoA analysis using the Bray–Curtis similarity metric revealed that the samples clustered according to stage and anatomical site (Figure 4A). This was also evidenced via unweighted pair–group method with arithmetic means (UPGMA), which measured significantly shorter branch lengths according to stage and anatomical site (Figure 4B, Appendix A). Interestingly, there was no separation between the rumen and cecum samples of the fetus, or between the AF and UCB groups (Figure 4A–C).

In the rumen, there were four sub-clusters, which consisted of the fetus groups (AF, FR and UCB groups), the newborn group (NBR), the drinking milk group (LR1d), and the after weaning groups (LR3m, LR6m and GR), while in the cecum data, the fetus groups and the newborn group were clustered together (Figure 4D,E). This indicated that the bacterial communities in the 3-month-old kids, 6-month-old kids and adult goats displayed a high degree of similarity, and suggested that drinking milk played an important role in the fluctuation of the microbiome.

A total of 27 phyla were identified from all samples (Appendix A). The majority of the sequences belonged to the Firmicutes, Bacteroidetes and Proteobacteria (Figure 5, Appendix A). However, the proportion of these phyla throughout the stages was different. Proteobacteria were mainly present in the fetus, newborn, one-day-old kid, UCB and AF. Bacteroidetes and Firmicutes were the two major phyla in the 3-month-old kids, 6-month-old kids and adult goats. In the rumen, the phylum Proteobacteria was found in samples taken from the FR, NBR and LR1d groups at significantly higher levels than in the three older age groups (*p* < 0.01), while there was a decreasing trend with increasing age in the cecum (*p* < 0.05). Conversely, the phylum Bacteroidetes showed the opposite trend. Interesting, the phylum Firmicutes showed no significant differences among the rumen samples (*p* > 0.05); however, in the cecum, Firmicutes became the most abundant phylum in samples from kids. The abundance of the phylum Spirochaetes in rumen samples taken from the fetus was significantly lower than that of their mothers (*p* < 0.01). In cecum, higher levels of Spirochaetes were found in samples taken from the LC6m and GC groups than in the other groups (*p* < 0.01). In particular, the phylum Fusobacteria was only found in the LR1d group. In addition, several minor phyla were noted, such as Tenericutes, Actinobacteria, Synergistetes and Verrucomicrobia, which could be found in all stages along the rumen and cecum.

At the genus level, a total of 3900 taxa were identified. Venn diagram analysis identified 228 taxa that were shared between the fetuses and their mothers, while 26 and 5 taxa were found to be shared between the rumen and cecum among the stages, respectively (Figure 6A). The abundant genera (those with an average proportion >10% in a group) are shown in Figure 6B and Appendix A. In the rumen and cecum, Unclassified_Comamonadaceae and Unclassified_Burkholderiales were the main fetal genera (average proportion > 18%). Interestingly, they were also the main genera in the UCB and AF samples. After birth, the relative abundances of these two genera decreased to 2.58% and 3.13% in the rumen, respectively, while they remained at high levels in the cecum. After 24 h of being breastfed, these genera decreased to below 1% in the rumen, and were undetectable in the cecum. Moreover, they were not detected in the later stages. The genera Bacteroides and Unclassified_Enterobacteriaceae were predominant in samples from the LC1d group compared with their levels in the other groups, reaching up to 37.53% and 28.20%, respectively. These two genera were also highly abundant in the LR1d group, representing 35.80% and 56.80% of the microbiota of one kid, respectively; however, their proportions in the other three kids were below 8%. The genera Bibersteinia, Bergeriella and Prevotella showed high levels in the LR1d group, while in the LC1d group the genus Butyricicoccus was highly abundant. In the 3-month-old kids, 6-month-old kids and adult goats, the genus Prevotella showed the highest abundance in the rumen, followed by Unclassified_BS11 and Unclassified_S24-7; however, in the cecum, Unclassified_Ruminococcaceae, Unclassified_Bacteroidales and Unclassified_Clostridiales were the main genera.

### 3.4. Rumen and Cecum Microbiomes of Nine Female Goats and Their Fetuses during Late Pregnancy

The diversity (Figure 3, Appendix A), structure (Figure 4) and composition (Figure 5 and Figure 6, Appendix A) of the microbial communities were distinct between pregnant goats and their fetuses. The diversity of the fetal rumen and cecum microbiomes was significantly lower (*p* < 0.05) than their maternal microbiomes (Figure 3, Appendix A). PCoA clustering analysis showed that neither the rumen nor the cecum microbiome from fetuses consistently resembled those of their mothers, and the rumen microbiome was distinguishable from the cecum microbiome in pregnant goats (Figure 4). This was also evidenced via ANOSIM, which yielded extremely significant (*p* < 0.001) and high R values (>0.8) between pregnant goats and their fetuses (Appendix A). Interestingly, the fetal rumen and cecum microbiomes from three pregnancy periods clearly clustered together, while the rumen microbiome clustered slightly differently from the cecum microbiome, suggesting that the fetal microbiota is highly conserved during late pregnancy, despite inter-individual variation. Furthermore, the proportions of the microbial communities of the fetuses and their mothers were clearly distinct at the phylum and genus levels, although they had mostly the same microbiomes (Figure 5 and Figure 6, Appendix A). For example, the most abundant phyla in the rumen and cecum microbiomes from the fetuses and their mothers were Proteobacteria, Firmicutes and Bacteroidetes, followed by Spirochaetes, Tenericutes, Euryarchaeota, Actinobacteria and Synergistetes (Figure 5, Appendix A). However, the proportions of these phyla in the rumen and cecum of fetuses and their mothers were different. Fetal communities were dominated by species from the Proteobacteria (57.70% in rumen, 59.17% in cecum), Bacteroidetes (12.81% in rumen, 7.51% in cecum) and Firmicutes (12.11% in rumen, 8.30% in cecum) phyla, which in part were similar to the embryonic microbiota in chickens [13]. Meanwhile, their mothers had a higher proportion of Bacteroidetes (65.78% in rumen, 52.67% in cecum) and Firmicutes (22.97% in rumen, 36.34% in cecum), as expected for the healthy adult goat and sheep [30,31]. At the genus level (Figure 6, Appendix A), Unclassified_Comamonadaceae (about 32%), Unclassified_Burkholderiales (about 18%) and Prevotella (3%) were the dominant microbes in the fetal rumen and cecum, while Prevotella (24.00%), Unclassified_Bacteroidales (14.13%), Unclassified_BS11 (13.85%) and Unclassified_S24-7 (5.98%) were the most abundant microbes in their mother’s rumen, and Unclassified_Bacteroidales (21.98%), Unclassified_Ruminococcaceae (17.08%), 5-7N15 (12.36%) and Unclassified_Clostridiales (10.42%) were dominant in their mother’s cecum. A total of 1196 OTUs were found in the rumen microbiomes of both the fetuses and their mothers, accounting for 30.97% and 60.07% of the sequences in the FR and GR groups, respectively. Moreover, 992 OTUs were found in the cecum, accounting for 31.31% and 48.87% of the sequences in the FC and GC groups, respectively (Figure 6A).

### 3.5. Predicted Function of Bacteria in Different Developmental Stages

This study used PICRUSt (Phylogenetic Investigation of Communities by Reconstruction of Unobserved States) as a predictive exploratory tool to predict the functional gene pathways in all 106 samples based on sequenced bacterial genomes. At KEGG level 2, a total of 41 gene families were identified in all samples (Figure 7A, Appendix A). According to the relative abundances of the KEGG pathways of the bacteria in the samples, PCoA showed a clear distinction between the clustering of fetus samples that of and other samples (Figure 7B). In these 41 gene families, the majority of genes belonged to membrane transport (11.10% of total genes inferred by PICRUSt), amino acid metabolism (10.15%), carbohydrate metabolism (9.97%), replication and repair (8.38%), energy metabolism (5.85%) and translation (5.30%). Of these six main gene families, either in the rumen or cecum, the proportion of the gene families associated with membrane transport decreased with age. This result correlated with the conventional metabolic functions that are indispensable for microbial subsistence [28], and the similar microbial potential functions of the goat gastrointestinal tract during preweaning development [9]. The KEGG level 3 showed that 307 KEGG ortholog (KO) pathways were present in all samples. Twenty-six prominent pathways (relative abundance ≥1%) were selected among these KO pathways, and the majority of genes belonged to transcription (4 KOs), membrane transport (3 KOs), replication and repair (3 KOs) and energy metabolism (3 KOs) (Appendix A).

## 4. Discussion

The rumen and cecum are two important fermentation organs in ruminants. The rumen converts indigestible feed plant mass into food products, such as milk and meat, and this process is mediated mainly by microbial degradation and fermentation. The cecum is where most of the chyme that has not been fermented completely in the rumen is fermented to produce final metabolites. Therefore, it is important to study the acquisition and development of the neonatal microbiome in the rumen and cecum, as well as the difference between these two fermentation organs. The results showed that the diversity (Figure 3, Appendix A), structure (Figure 4) and composition (Figure 5 and Figure 6, Appendix A) of the microbial communities were distinct between the rumen and cecum after birth, while they were similar in the fetal period (Figure 4). Bacterial diversity and richness represented a dynamic fluctuation of the microbiome (Figure 3A). The low diversity and richness of the microbiomes of one-day-old kids reflected the rapid loss and influx of microbes after receiving breast milk, and is partly consistent with a previous report that infants receiving some breast milk had significantly lower bacterial diversity when compared with infants no longer receiving breast milk [32]. After weaning, bacterial richness and diversity increased in each cluster, and they clustered together according to the PCoA clustering analysis (Figure 2B). There were three sub-clusters in both the rumen and cecum, which consisted of the fetus and newborn groups, the 1-day-old group and 3-month-old groups, and the 6-month-old group, compared with the pregnant goat group (Figure 4, Appendix A). In the rumen, the fetuses and newborns were not clustered as closely as they were in the cecum. This indicated that the bacterial communities in the fetus and newborn groups displayed a high degree of similarity, especially in the cecum, as did the 3-month-old and 6-month-old groups compared with the pregnant goat group.

At the phylum level, a substantial decrease in the phylum Proteobacteria and a rise in the phylum Bacteroidetes were observed in both the rumen and cecum (Figure 5). This latter change was caused by weaning, especially in the rumen. A similar compositional change in the Proteobacteria and Bacteroidetes phyla was reported in a study comparing the bovine rumen bacterial community from birth to adulthood [6]. Similar observations have been reported for goats during preweaning development in the cecum [9]. Bacteroidetes possess a strong ability to degrade proteins and polysaccharides [33,34], and these could indicate that the rumen is the place where nutrient digestion and metabolism mostly occur. There was no significant difference between the fetus and newborns at the phylum level; however, this began to change after the kids received breast milk, as indicated by the dynamic fluctuation of Proteobacteria, Bacteroidetes, Firmicutes and Fusobacteria (Figure 5). Furthermore, the compositions of the microbial communities at the phylum and genus levels changed radically after weaning, and the microbiomes of the two phases (3-month-old and 6-month-old) were similar to those of the pregnant goats (Figure 5), suggesting that microbiome stabilization occurred from month 3 of life.

At the genus level, we observed that some microbes were easily lost or replaced. The dominance of Unclassified_Comamonadaceae and Unclassified_Burkholderiales in the fetus decreased drastically in the rumen after birth (the relative abundance of Unclassified_Comamonadaceae dropped from 32.08% to 2.58%, and that of Unclassified_Burkholderiales decreased from 19.89% to 3.13%), while they only slightly decreased in the cecum (Unclassified_Comamonadaceae from 29.83% to 16.08%, Unclassified_Burkholderiales from 18.17% to 16.28%) (Figure 6A, Appendix A). After receiving breast milk, these two dominant bacteria drastically decreased to 0% in both the rumen and cecum, and were not detected in later phases (Figure 6A, Appendix A). One explanation was provided by Ferretti et al., who stated that some microbes in stool were present only transiently in some infants because they are probably poorly adapted or unsuited to colonizing the infant’s lower gastrointestinal tract [16]. By contrast, Mannheimia and Bibersteinia were rapidly acquired in the rumen but not in the cecum after birth, despite inter-individual variation. These two dominant bacteria, Mannheimia (18.33%) and Bibersteinia (13.63%), are pathogenic microorganisms. Mannheimia haemolytica and Bibersteinia trehalosi are the main causes of pneumonia in sheep and goats of all ages, and can induce septicemia in lambs or kids [35,36]. The results revealed that kids born in a natural environment without quarantine might easily acquire pathogenic microorganisms. Interestingly, Mannheimia was not present in the rumen after the kids received breast milk, while Bibersteinia was retained at relatively high abundance (10.73%) until weaning, demonstrating that there were some antibodies and immune factors in the colostrum that could resist Mannheimia, and some factors in the diet that could resist Bibersteinia. In the cecum, the relative abundances of Bacteroides and Unclassified_Enterobacteriaceae rapidly increased to 37.53% and 28.20% after receiving breast milk, respectively, and these two genera were the dominant bacteria in the LC1d group. However, they were either lost or were undetectable after weaning. In particular, Butyricicoccus was the dominant bacteria (a relative abundance of 10.55%) and was only detected in the LC1d group, which was consistent with it being mainly present in the large intestine in kids during preweaning development [9]. Butyricicoccus is a butyrate-producing clostridial cluster IV genus whose numbers are reduced in the stool of patients with ulcerative colitis, and many studies have confirmed that Butyricicoccus pullicaecorum could reduce intestinal inflammation [37,38,39,40]. This finding suggested that the hircine colostrum could be used to treat intestinal inflammation. Thus, the microbiomes of the goat gastrointestinal tract from different periods were distinct, and they were closely associated with the environment and diet. The predominant genera in the cecal microbiota were unclassified_Ruminococcaceae, unclassified_Clostridiales and unclassified_Bacteroidales from 3 months of life (Figure 6B, Appendix A). Prevotella, which was the main genus under the phylum Bacteroidetes, showed high levels (the relative abundance > 10%) in the rumen, while it was low (<1%) in the cecum from 3 months of life (Figure 6B, Appendix A). These findings agreed with those of previous studies [7,30]. The reason that unclassified Ruminococcaceae and unclassified Clostridiales were enriched in the intestine is not clear yet [30]. The genus Prevotella was found to degrade nonstructural carbohydrates and proteins, as well as be involved in amino acid metabolism, nucleotide metabolism, energy metabolism and glycan biosynthesis [30,41,42,43].

In the present study, we hypothesized that fetuses developed in a nonsterile environment because microorganisms isolated from the meconium of healthy newborns by vaginal delivery or cesarean section indicated that their microbiota was not derived only postnatally [14,15,16,17]. For instance, many cultivable microorganisms were present in the umbilical cord blood of preterm infants, and abundant microorganisms have been detected in amniotic fluid [44,45,46,47]. Accordingly, microbes might be present in the fetus. For example, the fetus was enriched with microbial species present in the microbiomes of the umbilical cord blood and amniotic fluid, suggesting that the possible origin of the neonatal microbiomes was during the fetal period, which could explain why the microbiomes of newborn meconium did not consistently resemble one specific maternal body site, such as the oral cavity, skin, vagina, or the breast milk [16].

We acknowledge several limitations of our study. For example, some microbial species from the sampling process, DNA extraction and PCR reagents were unavoidably present in the samples, even though we have removed many microbial species. On the other hand, the methods used in this study allow only for the detection of bacterial DNA, and not viability. Thus, more precise quantification of contamination using other analysis methods are needed to confirm our results. Besides that, there were very low sample sizes (*n* = 3) in some groups, such as in three-month-olds after weaning (LR3m and LC3m).

## 5. Conclusions

In summary, the diversity, structure and composition of the goat microbial communities were distinct between the rumen and cecum after birth, while they were similar in the fetal period. Importantly, the dominant bacteria of the newborns were rapidly lost or were present at undetectable levels after drinking colostrum, which might represent the second phase of the microbiome of the gastrointestinal tract, with the third phase beginning after weaning. The results of the present study revealed the microbial transmission routes from the mother to fetus, and also revealed the establishment and dynamic fluctuations of the gut microbiome from fetus to adulthood in goats.

## Figures and Tables

**Figure 1 animals-10-01639-f001:**
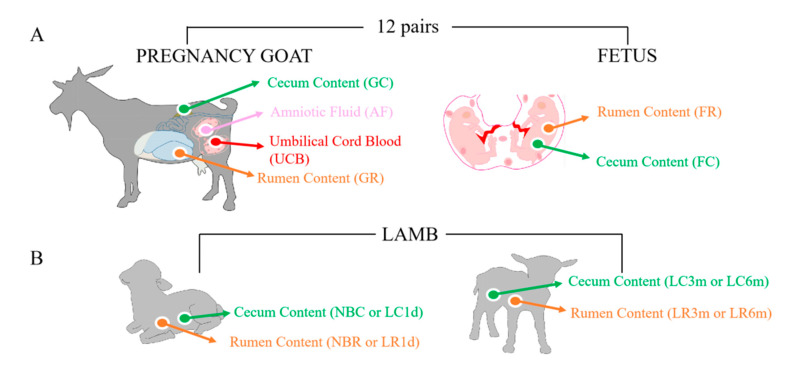
16S rRNA gene sequencing of the microbiome of mother–fetus pairs and kids. (**A**) Samples were collected from nine mother–fetus pairs. Samples were taken from the rumen fluid (GR), cecum content (GC), umbilical cord blood (UCB) and amniotic fluid (AF) of the mothers and from the rumen fluid (FR) and cecum content (FC) of the fetus. (**B**) Samples were collected from 16 kids. Samples were taken from the rumen fluid (R) and cecum content (C) from newborns without receiving breast milk (NBR and NBC), one-day-old kids after receiving breast milk (LR1d and LC1d), three-month-old kids after weaning (LR3m and LC3m), and six-month-old kids (LR6m and LC6m). All samples were 16S rRNA gene sequenced and the average depth (in Gbases) of the quality-controlled reads was determined.

**Figure 2 animals-10-01639-f002:**
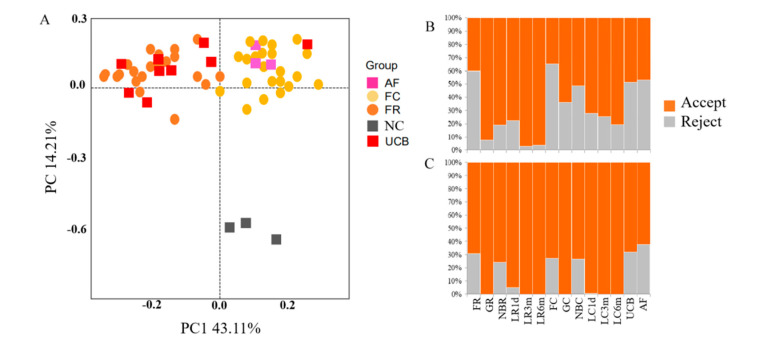
Contamination control and removal of potential contaminant sequences. (**A**) Principal coordinate analysis (PCoA) at the OTU level of the community structure in bacteria of fetal rumen fluid (FR) and cecum content (FC), umbilical cord blood (UCB), amniotic fluid content (AFC) and the negative control (NC). (**B**) Effect of data rejection on the animal microbiota samples. Each bar represents all sequence reads obtained from one sample. Orange shows the proportion of OTUs (de-noised 16S rDNA sequences) found only in the actual samples. Gray shows OTUs that were shared between samples and negative controls. (**C**) Orange shows the proportion of OTUs that were accepted in samples. Gray shows sequences that were removed because of their abundance in the negative controls, the amounts of said sequences in published material on contamination, and the major microbes in the healthy adult goat.

**Figure 3 animals-10-01639-f003:**
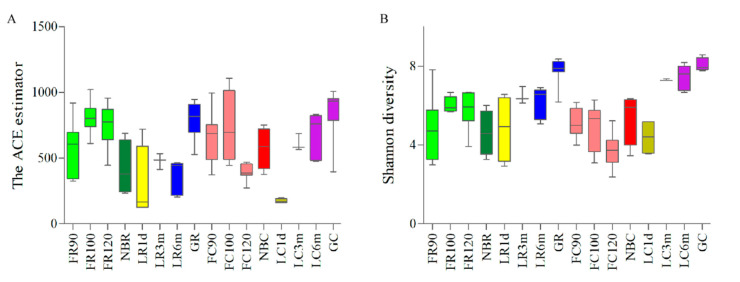
Box plots showing the alpha diversity per stage in the rumen and cecum. The richness and diversity were calculated via ACE (**A**) and Shannon (**B**) index. Samples were taken from the rumen fluid (GR) and cecum content (GC) of the mothers and from the rumen fluid (FR) and cecum content (FC) of the fetus, and the fetus were at three time points: 90, 100 and 120 days. In addition, samples were taken from the rumen fluid (R) and cecum content (C) of 16 kids: newborns without receiving breast milk (NBR and NBC), one-day-olds after receiving breast milk (LR1d and LC1d), three-month-olds after weaning (LR3m and LC3m) and six-month-old kids (LR6m and LC6m).

**Figure 4 animals-10-01639-f004:**
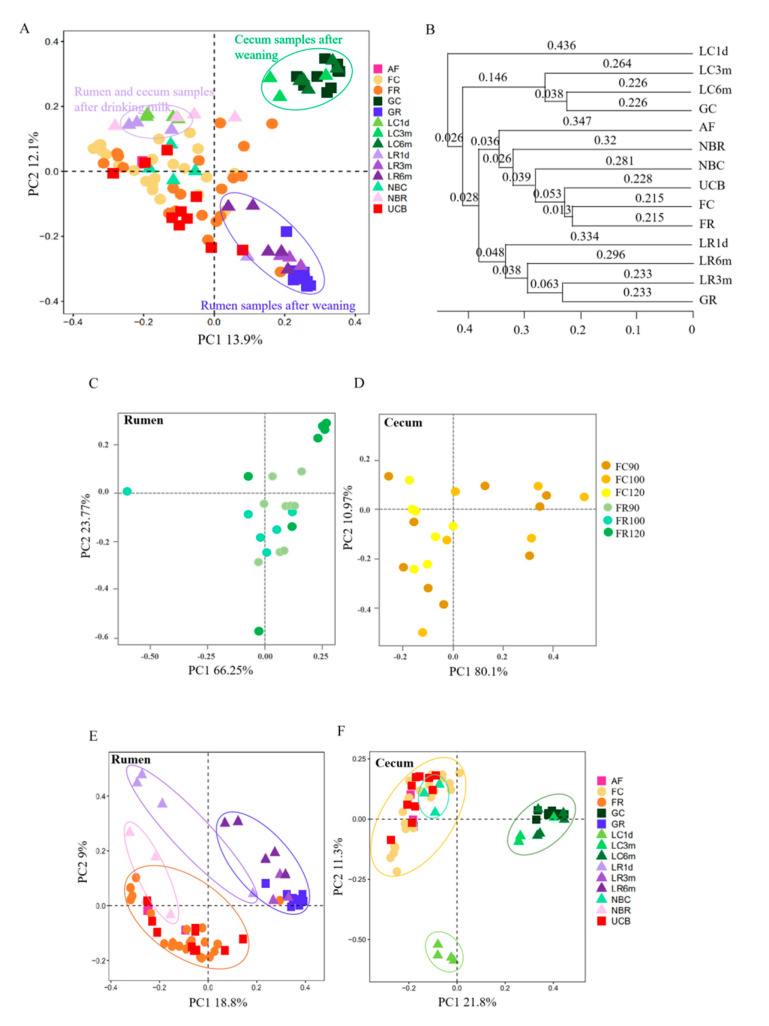
Principal coordinate analysis (PCoA) at the OTU level of the community structure in bacteria of mother–fetus pairs and kids with data decontamination on the animal microbiota samples. The abscissa represents the first principal component (PC1), the ordinate represents the second principal component (PC2), and the percentages in parentheses represent the contribution of the PC to the sample difference. (**A**) PCoA of bacterial microbiota across all samples. (**B**) Unweighted pair–group method with arithmetic means (UPGMA) of bacterial microbiota across all groups. (**C**,**D**) PCoA of the bacterial microbiota in the rumen (**C**) and cecum (**D**) samples of the fetus. (**E**,**F**) PCoA of the bacterial microbiota according to age group in each anatomical site.

**Figure 5 animals-10-01639-f005:**
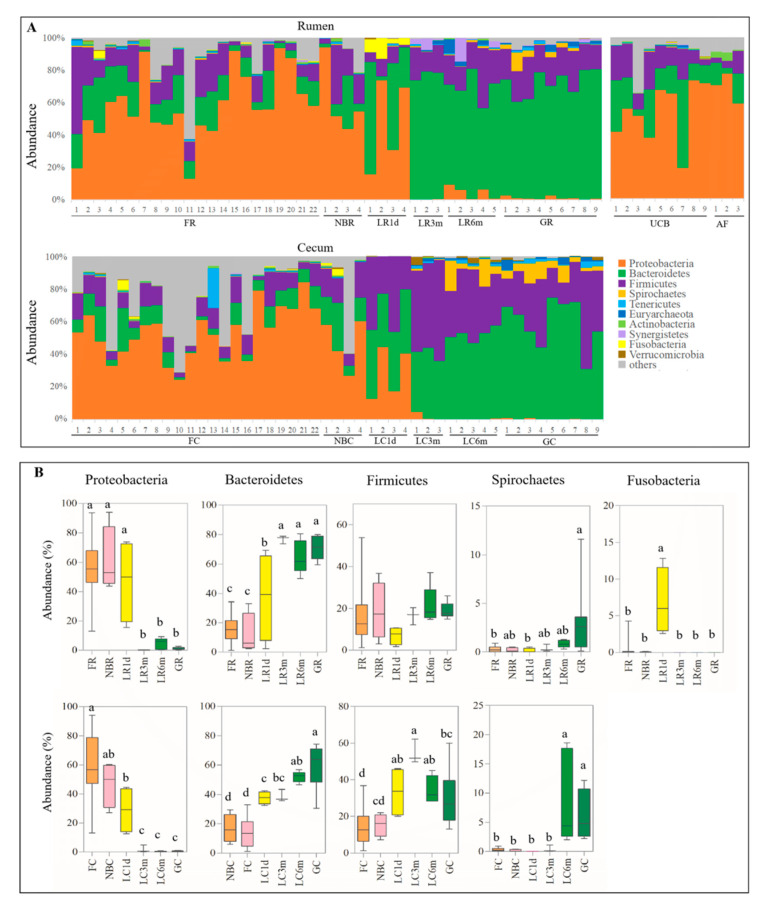
Microbiota composition at the phylum level. (**A**) Relative phylum-level abundance profiles for samples from the mother–fetus pairs and kids. (**B**) Comparison of the relative abundances of the main bacterial phyla (Firmicutes, Bacteroidetes, Proteobacteria, Spirochaetes and Fusobacteria) among the different stages in the rumen and cecum.

**Figure 6 animals-10-01639-f006:**
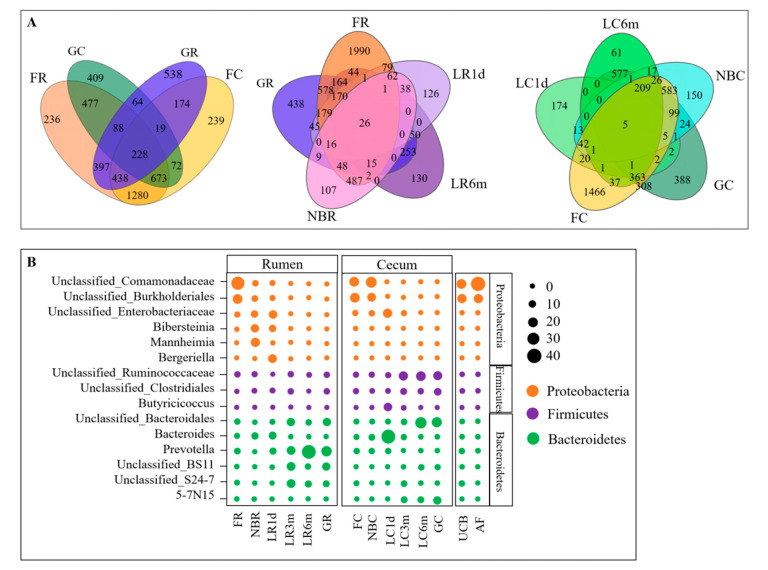
Main common genera in each group. (**A**) Venn diagrams demonstrating the distribution of the OTUs shared among all groups. The OTU numbers were generated from subsets of each group. (**B**) Relative genus-level abundance profiles for samples from the mother–fetus pairs and kids (those with an average proportion >10% in a group are shown).

**Figure 7 animals-10-01639-f007:**
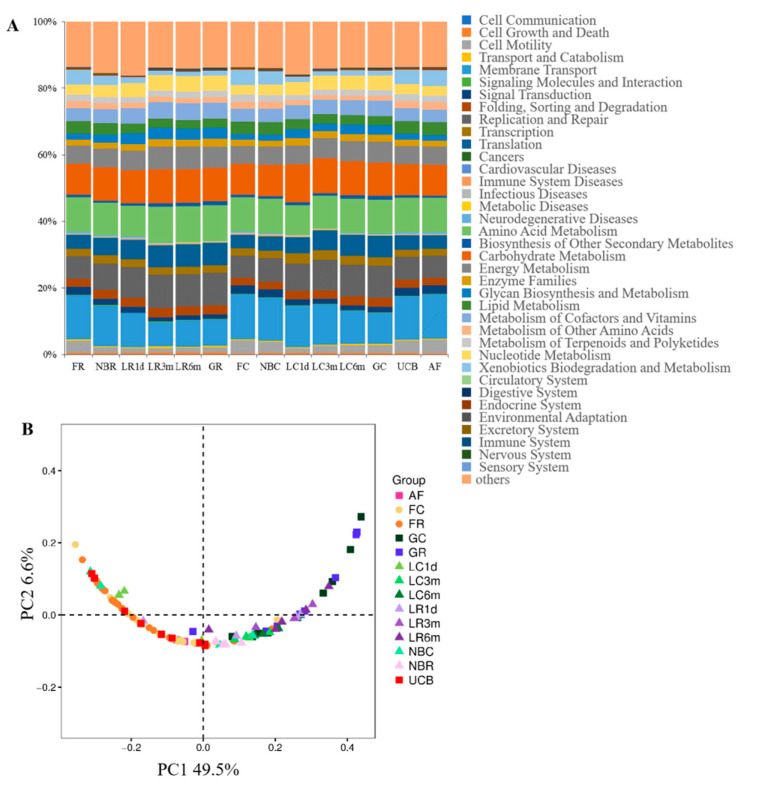
Metagenomic functional predictions. (**A**) Variations in KEGG metabolic pathways of microbiota at different ages between rumen and cecum. (**B**) PCoA profile of microbial functional KEGG pathways using the Bray–Curtis dissimilarity metric according to the abundance of metabolic pathways.

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
