# Peer review of "Exploring the Rumen and Cecum Microbial Community from Fetus to Adulthood in Goat"

_animals, 2020, doi:10.3390/ani10091639_

Round 1

Reviewer 1 Report

Dear authors,

compliment for the paper, very well done. I could not find parts for improvement and on the other hand: this can be published how it is now. Congratulations.

Author Response

compliment for the paper, very well done. I could not find parts for improvement and on the other hand: this can be published how it is now. Congratulations.

Response: We do appreciate your positive affirmation of the work.

Reviewer 2 Report

Dear editor and authors,

This is an interesting study evaluating the colonization of the intestinal tract of goat kids. As authors mentioned in the introduction, the uterine microbiome is quite controversial, and although negative control samples for PCR reagents were included, there were no negative controls for sampling, which is crucial in this type of study. In many cases, half or more of reads were originated in the PCR or sequencing reagents. It is possible that the other half were present during sampling, but you cannot prove this is not true because of the lack of controls. All the material used for collection were sterile, but not DNA free, so it is possible/likely that there were traces of bacterial DNA. Indeed, the similarity between all samples collected with this procedure (fetal rumen, cecum, umbilical cord) could be because of contamination. Furthermore, the methods used in this study allow only for detection of bacterial DNA, and not viability.

Your data is interesting and worthy publication, but you methods do not allow you to state that there was a fetus microbiota especially considering the controversial nature of this topic. Thus, at this point, conclusions of the study cannot be supported by the results because of this major flaw in the study design.

Besides that, another limitation is the inclusion of too many variables, resulting in some groups with very low sample sizes (n=3). Other variables potentially affecting the microbiota were not even considered for the analysis, such as slaughtering at 2 places and at 3 different times, presence of duplets and triplets.

It is not clear if pregnant goats were anesthetized and fetus extracted and then euthanized? Samples collected at the slaughter house or university?

Line 101: what do you mean by “fetus”? Where in the fetus?

Line 102: within 20 min of what?

Line 126: what did you use QIIME for?

Line 137: which negative controls? From PCR?

Figure 2: is the PCoA really evaluating bacterial diversity? Be careful with the nomenclature you use because diversity is related to alpha diversity.

Figure 3: legend should explain what manes in x axis mean.

The abbreviations are not intuitive.

Line 270: statistically different?

Figure 6: because you have so many groups some number overlap and is quite impossible to understand the sharing between groups.

Line 314: As I mentioned earlier, I think this whole analysis is inappropriate, considering your study design.

You acknowledge “many limitations of the study” but mention only one.

Author Response

Response to Reviewer 2 Comments

  1. This is an interesting study evaluating the colonization of the intestinal tract of goat kids. As authors mentioned in the introduction, the uterine microbiome is quite controversial, and although negative control samples for PCR reagents were included, there were no negative controls for sampling, which is crucial in this type of study. In many cases, half or more of reads were originated in the PCR or sequencing reagents. It is possible that the other half were present during sampling, but you cannot prove this is not true because of the lack of controls. All the material used for collection were sterile, but not DNA free, so it is possible/likely that there were traces of bacterial DNA. Indeed, the similarity between all samples collected with this procedure (fetal rumen, cecum, umbilical cord) could be because of contamination. Furthermore, the methods used in this study allow only for detection of bacterial DNA, and not viability.

Response 1: Yes, we agree. This was another limitation of our work. We added the sentence “On the other hand, the methods used in this study allow only for detection of bacterial DNA, and not viability.” in lines 480-481.

  1. Your data is interesting and worthy publication, but you methods do not allow you to state that there was a fetus microbiota especially considering the controversial nature of this topic. Thus, at this point, conclusions of the study cannot be supported by the results because of this major flaw in the study design.

Response 2: Thank you for the good comment. We deleted the sentences “The results supported the view that microbes present in the fetus were acquired and colonized from umbilical cord blood” to the conclusions in line 487, and change “fetus” to “newborn” in line 488. Finally, the conclusions were changed to “In summary, the diversity, structure, and composition of the goat microbial communities were distinct between the rumen and cecum after birth, while they were similar in the fetal period.  Importantly, the dominant bacteria of the newborn were rapidly lost or were present at undetectable levels after drinking colostrum, which might represent the second phase of the microbiome of the gastrointestinal tract, with the third phase beginning after weaning. The results of the present study revealed the microbial transmission routes from the mother to fetus, and revealed the establishment and dynamic fluctuations of the gut microbiome from fetus to adulthood in goats.” in lines 486-493.

  1. Besides that, another limitation is the inclusion of too many variables, resulting in some groups with very low sample sizes (n=3). Other variables potentially affecting the microbiota were not even considered for the analysis, such as slaughtering at 2 places and at 3 different times, presence of duplets and triplets.

Response 3: Thanks for the comment. We added the sentence “Besides that, there were very low sample sizes (n=3) in some groups, such as three months old after weaning (LR3m and LC3m).” in lines 483-484.

  1. It is not clear if pregnant goats were anesthetized and fetus extracted and then euthanized?

Response 4: Yeah, exactly. We added the sentence “The pregnant goats were anesthetized and fetus extracted and then euthanized.” in lines 99-100.  

  1. Samples collected at the slaughter house or university?

Response 5: We added the sentence “and samples were collected at the slaughter house” in line 105.

  1. Line 101: what do you mean by “fetus”? Where in the fetus?

Response 6: Thank you for the good advice. We were sorry we made a mistake. We deleted “, and fetus”, and changed the sentence to “Samples of umbilical cord blood, amniotic fluid were collected from pregnant goats in 20 minutes. ” in line 113.

  1. Line 102: within 20 min of what?

Response 7: We changed “within 20 min” to “in 20 minutes” in line 113.

  1. Line 126: what did you use QIIME for?

Response 8: QIIME is an open source software for comparing and analyzing the microbiome. The official website http://qiime.org/. Please see the reference 23.

  1. Line 137: which negative controls? From PCR?

Response 9: Yes, it is. We added the sentence “ (in the form of a PCRamplified kit blank sample)” in line 148.

  1. Figure 2: is the PCoA really evaluating bacterial diversity? Be careful with the nomenclature you use because diversity is related to alpha diversity.

Response 10: Thank you for the good comment. We changed the sentence  “Principal coordinate analysis (PCoA) profile of bacterial diversity from ...” to “Principal coordinate analysis (PCoA) at the OTU level of the community structure in bacteria of ...” in line 239 for Figure 2 and in lines 274-275 for Figure 4.

  1. Figure 3: legend should explain what manes in x axis mean.

Response 11: Yes, we agree. We added the sentence “Samples were taken from the rumen fluid (GR) and cecum content (GC) of the mothers and from the rumen fluid (FR) and cecum content (FC) of the fetus, and the fetus were at three time points: 90, 100, and 120 days, respectively. In addition, samples were taken from the rumen fluid (R) and cecum content  (C) of 16 kids: newborns without receiving breast milk (NBR and NBC), one day old after receiving breast milk (LR1d and LC1d), three months old after weaning (LR3m and LC3m), and six-month-old kids (LR6m and LC6m).” in lines 266-272.

  1. The abbreviations are not intuitive.

Response 12: Which abbreviation was not intuitive? Could you please point out?

  1. Line 270: statistically different?

Response 13: We though that the words “statistically different” were not the good description here. For more explanation, please see the following sentence in lines 290-303.

  1. Figure 6: because you have so many groups some number overlap and is quite impossible to understand the sharing between groups.

Response 14: Yes, we agree. Hence, we just paid attention on the number of taxa that were shared between the fetuses and their mothers, and between the rumen and cecum among the stages: Venn diagram analysis identified 228 taxa that were shared between the fetuses and their mothers, while 26 and 5 taxa were found to be shared between the rumen and cecum among the stages, respectively (Figure 6A). Please see in lines 310-312.

  1. Line 314: As I mentioned earlier, I think this whole analysis is inappropriate, considering your study design.

Response 15: The origin or seed of the detected fetal microbial community was an open debate. To avoid causing a great deal of controversy, we deleted these part (3.4. Early Acquisition and Characteristics of the Fetal Microbiome) after careful consideration.

Accordingly, we deleted the sentence “which might have been acquired and colonized from umbilical cord blood,” to the Abstract in lines 39-40.

  1. You acknowledge “many limitations of the study” but mention only one.

Response 16: We added two sentence to the conclusions: “On the other hand, the methods used in this study allow only for detection of bacterial DNA, and not viability” in lines 480-481, and “Besides that, there were very low sample sizes (n=3) in some groups, such as three months old after weaning (LR3m and LC3m)” in lines 483-484.

Reviewer 3 Report

The manuscript shows a good effort by authors in giving some insight in the development of both caecum and rumen microbiota from before born until adulthood. This effort has to be aknowledge, since it is an ambicious work. There are still some general aspects that should be refined and amended prior its final acceptance but the study itself has its merit.

In terms of formal aspects, there are some citations that have been duplicated in the reference list such as number 9 and 10, or 11 and 35. This shows a poor self review by the authors, and I encourage to be more careful in the future since that is not a reviewer's task to be completed.

Going into the matter, I find both introduction and discussion poor. The work itself has a huge potential but it looks authors stay in the surface. There hasn´t been discussed properly the relative importance of the main microbial groups for the animal metabolism, and how the changes that are observed, modify that metabolism, it just merely describes the process without going deeper. My suggestion is to improve both sections.

In methods/results, I would have appreciated some information related with animal diets (chemical composition of calostrum, milk, and diet in adults as well as list of ingredients when appropriate) and fermentation parameters in rumen and caecum samples (SCFA mainly, but maybe also lactate, ammonia or pH). That supporting information is lacking and it would have helped to support discussion section.

Number of animals is short but the design is deep enough to get some conclusions although they may have to use them with care.

I won´t go into specific correcions in both introduction and discussion since I expect a thorough reading and amendment in both sections by the authors, but there are some errors it will need to be fixed eventually.

Author Response

Response to Reviewer 3 Comments

  1. The manuscript shows a good effort by authors in giving some insight in the development of both caecum and rumen microbiota from before born until adulthood. This effort has to be aknowledge, since it is an ambicious work. There are still some general aspects that should be refined and amended prior its final acceptance but the study itself has its merit.

Response 1: We do appreciate your positive affirmation of the work.

  1. In terms of formal aspects, there are some citations that have been duplicated in the reference list such as number 9 and 10, or 11 and 35. This shows a poor self review by the authors, and I encourage to be more careful in the future since that is not a reviewer's task to be completed.

Response 2: Thank you for your kind advice. Sorry we make a mistake. We had deleted the duplicated references, such as number 10 (duplicated with number 9), 34 (28), 35 (11), 38 (16), please see in the reference list.

In addition, we modified the Introduction and Discussion part, and added some references. Hence, we changed the reference list.

  1. Going into the matter, I find both introduction and discussion poor. The work itself has a huge potential but it looks authors stay in the surface. There hasn´t been discussed properly the relative importance of the main microbial groups for the animal metabolism, and how the changes that are observed, modify that metabolism, it just merely describes the process without going deeper. My suggestion is to improve both sections.

Response 3: Yes, we agree. We added many sentence to Introduction and re-wrote the Discussion.

  1. In methods/results, I would have appreciated some information related with animal diets (chemical composition of calostrum, milk, and diet in adults as well as list of ingredients when appropriate) and fermentation parameters in rumen and caecum samples (SCFA mainly, but maybe also lactate, ammonia or pH). That supporting information is lacking and it would have helped to support discussion section.

Response 4: Thank you for the good comment. We added the sentence “All the goats received freely available water and were fed total mixed ration (TMR) (purchased from Wens Foodstuff Group Co., Ltd.) throughout the experiment.” to the Methods in lines 89-91. We are sorry that we didn’t do the research of SCFA , lactate, ammonia or pH.   

  1. Number of animals is short but the design is deep enough to get some conclusions although they may have to use them with care.

Response 5: Thank you for the suggestion. 

  1. I won´t go into specific correcions in both introduction and discussion since I expect a thorough reading and amendment in both sections by the authors, but there are some errors it will need to be fixed eventually.

Response 6: Done as the mention earlier. We added many sentence to Introduction and re-wrote the Discussion.

This manuscript is a resubmission of an earlier submission. The following is a list of the peer review reports and author responses from that submission.